# Impact of a High-Fat Diet on the Metabolomics Profile of 129S6 and C57BL6 Mouse Strains

**DOI:** 10.3390/ijms231911682

**Published:** 2022-10-02

**Authors:** Maria Piirsalu, Egon Taalberg, Mohan Jayaram, Kersti Lilleväli, Mihkel Zilmer, Eero Vasar

**Affiliations:** 1Institute of Biomedicine and Translational Medicine, Department of Physiology, University of Tartu, 19 Ravila Street, 50411 Tartu, Estonia; 2Center of Excellence for Genomics and Translational Medicine, University of Tartu, 50411 Tartu, Estonia; 3Institute of Biomedicine and Translational Medicine, Department of Biochemistry, University of Tartu, 19 Ravila Street, 50411 Tartu, Estonia

**Keywords:** 129S6/SvEvTac, C57BL/6NTac, metabolomics, standard diet, high-fat diet

## Abstract

Different inbred mouse strains vary substantially in their behavior and metabolic phenotype under physiological and pathological conditions. The purpose of this study was to extend the knowledge of distinct coping strategies under challenging events in two differently adapting mouse strains: C57BL/6NTac (Bl6) and 129S6/SvEvTac (129Sv). Thus, we aimed to investigate possible similarities and differences in the body weight change, behavior, and several metabolic variables in Bl6 and 129Sv strains in response to high-fat diet (HFD) using the AbsoluteIDQ p180 kit. We found that 9 weeks of HFD induced a significant body weight gain in 129Sv, but not in Bl6 mice. Besides that, 129Sv mice displayed anxiety-like behavior in the open-field test. Metabolite profiling revealed that 129Sv mice had higher levels of circulating branched-chain amino acids, which were even more amplified by HFD. HFD also induced a decrease in glycine, spermidine, and t4-OH-proline levels in 129Sv mice. Although acylcarnitines (ACs) dominated in baseline conditions in 129Sv strain, this strain had a significantly stronger AC-reducing effect of HFD. Moreover, 129Sv mice had higher levels of lipids in baseline conditions, but HFD caused more pronounced alterations in lipid profile in Bl6 mice. Taken together, our results show that the Bl6 line is better adapted to abundant fat intake.

## 1. Introduction

C57BL/6NTac (Bl6) and 129S6/SvEvTac (129Sv) are among the most widely used mouse strains in biomedical and transgenic research and are the gold standard for creating transgenic mouse models. Given the rising usage of genetically modified mice, it is becoming increasingly important to consider genetic background when modelling human diseases, as it may have a significant impact on the outcome. Thus, it is crucial to gain new insight into strain-specific variations in order to improve our comprehension of proper model selection [1].

Bl6 and 129Sv mice are substantially different from one another in many aspects. Bl6 mice actively cope in stressful situations, whereas the responding strategy of the 129Sv line is inherently passive [2]. Use of prior environmental enrichment amplifies the exploratory activity of Bl6 strain in a novel and stressful environment [3], whereas in similar conditions, 129Sv strain display increased anxiety and body weight loss [3,4]. A recent study confirms the vast difference of the behavior and body weight regulation in Bl6 and 129Sv strains [2]. In home-cage conditions, 129Sv mice gain significantly more body weight compared with Bl6 mice in the same environment. The opposite change was demonstrated in mice subjected to stressful repetitive motility testing for 11 days: 129Sv mice had a considerable reduction in body weight, while Bl6 mice showed essentially no change [2]. It is important to emphasize that in the motility test, the locomotor activity of Bl6 mice was much higher compared with 129Sv mice [2]. A notable feature was the steady and robust increase in rearings in the Bl6 mice, indicating active adaptation [2]. Taken together, 129Sv displays a greater discrepancy in body weight change if the outcomes of two interventions are compared. Similar results concerning dynamics of body weight were obtained in studies where these strains were exposed before to the enriched environment [4].

We have recently established that the metabolite signatures of Bl6 and 129Sv strains vary as well [2,5]. The metabolic signature of Bl6 contains three biogenic amines (acetyl-ornithine (Ac-Orn), alpha-aminoadipic acid (alpha-AAA), carnosine and lysophosphatidylcholine 16:1 (LysoPC 16:1). However, short-chain acylcarnitines (SCAC) C4- and C5- and sphingolipid SM(OH) C22:2 belong to the metabolic signature of 129Sv mice. Acylcarnitine (AC) C5- is a mixture of two isomers: isovalerylcarnitine and 2- methyl butyrylcarnitine. It has been demonstrated that the accumulation of C5- AC in 129Sv mice is due to mutation in the *lvd* gene resulting in isovaleryl-CoA dehydrogenase deficiency [6]. Alpha-AAA is an established biomarker for insulin resistance and diabetes risk [7,8,9]. It has been also demonstrated that Alpha-AAA is an inhibitor of kynurenic acid synthesis, which is a neuroactive metabolite and antagonist of the glutamatergic N-methyl-D-aspartate (NMDA) as well as AMPA/kainite and alpha 7 nicotinic receptors [10,11,12]. Higher levels of Alpha-AAA in Bl6 mice have been shown to be the consequence of a defect in the *Dhtkd1* gene [6,13]. Carnosine is a dipeptide (beta-alanyl-L-histidine) that is found in abundancies in excitable tissues including muscle and the brain that can scavenge free radicals and possess neuroprotective characteristics [14,15].

Moreover, we have recently demonstrated that systemic administration of lipopolysaccharide (LPS) causes hypometabolism in the Bl6 strain, which is advantageous for host tolerance, but promotes the production of proinflammatory metabolites in the 129Sv strain [5]. Several studies have also indicated that consumption of high-fat foods may induce systemic inflammation [16,17,18]. In this context, the aim of this study was to shed light on the impact of genetic background on the metabolic response to high-fat diet (HFD) in actively coping (Bl6) and passively coping (129Sv) mouse strains. Taking into account that 129Sv mice gain more body weight in home cages compared with Bl6 strain, one might expect similar effects with HFD. In addition, 129Sv mice are more sensitive to stress. Therefore, mice were also subjected to the open-field test (PhenoTyper test) at the beginning and end of the 9-week HFD feeding. To establish the HFD-induced alterations in metabolomics, blood samples were collected for metabolite analysis after the second open-field test using the AbsoluteIDQ p180 kit.

## 2. Results

### 2.1. Body Weight Dynamics, Food, and Water Intake

Bl6 and 129Sv mice were fed HFD or control diet (Figure 1K) for 9 weeks. Food and water consumption and body mass of each animal was recorded weekly. The initial body weight of all mice was approximately the same in all four groups. During the 9-week study period, the body weight gain of Bl6 mice fed with HFD did not differ from that of control diet. It means that weekly body weight dynamics of HFD-fed mice followed the same pattern as in control animals (Figure 1A). HFD-fed 129Sv mice began to weigh significantly more than control diet mice starting from the second week of dietary exposure (Figure 1D). 

Food and water consumption was measured by subtracting the mass (g) or volume (mL) of remaining food and water from the initial food and water mass or volume provided to each cage weekly and was corrected for the number of animals per cage. Food intake amount in HFD-fed 129Sv and Bl6 mice was significantly lower than in control chow-fed mice (Figure 1B,E). Water intake did not differ between HFD and control diet groups (Figure 1C,F).

Body weight and food and water intake was compared between strains by calculating the area under the curve (AUC). Differences among groups were assessed by two-way ANOVA (strain × diet) followed by Bonferroni post hoc test. For body weight, two-way ANOVA yielded a significant strain-by-diet interaction effect (*F*_(1, 54)_ = 6.21, *p* = 0.02). Comparison of groups demonstrated that AUC of body weight was significantly higher in HFD-fed 129Sv mice compared with 129Sv control mice (Figure 1G). The effect of food intake was also significant for strain (*F*_(1, 8)_ = 17.56, *p* = 0.003). Food intake of HFD-fed 129Sv mice was significantly lower compared with their control counterparts (Figure 1H). No significant difference was detected in water consumption between groups (Figure 1I).

Furthermore, two-way ANOVA demonstrated significant increase in 9-week body weight change (in %) between the HFD and control-diet-fed 129Sv mice (diet: *F*_(1, 54)_ = 23.45, *p* < 0.0001; strain: *F*_(1, 54)_ = 2.09, *p* = 0.15; strain × diet: *F*_(1, 54)_ = 3.81, *p* = 0.056). The 9-week weight gain of Bl6 mice fed with HFD was also slightly higher compared with their control counterparts; however, it did not reach statistical significance level (Figure 1J).

### 2.2. Impact of HFD on Locomotor Activity

Open-field testing was conducted at the beginning of dietary intervention and 9 weeks later. Locomotor activity was recorded for 24 h in PhenoTyper cages. The open-field arena was virtually divided into central, peripheral, and food zones (Figure 2Y). Total distance traveled in the whole arena and the time spent in the center and food zone were recorded and divided into light and dark cycles. Locomotor data was log2-transformed prior to repeated-measures ANOVA (time × diet) or two-way ANOVA (stain × diet) analyses to make the data correspond to normal distribution. 

We first examined whether there are differences between day 1 and week 9 open-field testing. Control-diet-fed Bl6 mice tended to spend less time in center and in the food zone on week 9 compared with day 1 (Figure 2B,C). HFD-fed Bl6 mice spent significantly less time in the food zone compared with their control counterparts on day 1 and on week 9 (time: (*F*_(1, 20)_ = 6.98, *p* = 0.02; diet: *F*_(1, 22)_ = 15.05, *p* = 0.0008; Figure 2C). Open-field activity of 129Sv mice was significantly reduced on week 9 compared with total distance traveled on day 1 in both control and HFD groups. During the 24 h cycle, HFD-fed 129Sv mice displayed a significant suppression of locomotor activity in the whole arena on day 1 and at week 9 (time: *F*_(1, 21)_ = 28.86, *p* < 0.0001; diet: *F*_(1, 21)_ = 19.81, *p* = 0.0002; Figure 2D). Interestingly, these HFD-fed mice displayed higher anxiety-like behavior as they spent significantly less time in the center zone compared with control-diet-fed 129Sv mice (time: *F*_(1, 21)_ = 11.05, *p* = 0.003; diet: *F*_(1, 21)_ = 13.19, *p* = 0.002; Figure 2E) on week 9.

Comparison of Bl6 and 129Sv strains revealed significant differences in total distance traveled and time spent in the center zone between HFD-fed mice on day 1 (Figure 2G,H) and on week 9 (Figure 2J,K). More precisely, HFD-fed Bl6 mice traveled a greater distance in the whole arena (day 1: strain: *F*_(1, 43)_ = 27.29, *p* < 0.0001, diet: *F*_(1, 43)_ = 5.46, *p* = 0.02; week 9: strain: *F*_(1, 43)_ = 58.73, *p* < 0.0001, diet: *F*_(1, 43)_ = 6.93, *p* = 0.01) as well as spending more time in the center zone (day 1: strain: *F*_(1, 43)_ = 13.48, *p* = 0.0007; week 9: strain: *F*_(1, 44)_ = 17.24, *p* = 0.0001, diet: *F*_(1, 44)_ = 6.95, *p* = 0.01, interaction: *F*_(1, 44)_ = 5.94, *p* = 0.02) compared with HFD-fed 129Sv mice. Bl6 control mice spent significantly more time in the food zone compared with 129Sv control mice on day 1 (strain: *F*_(1, 42)_ = 18.18, *p* = 0.0001, diet: *F*_(1, 42)_ = 15.93, *p* = 0.0003; Figure 2I) and on week 9 (strain: *F*_(1, 44)_ = 13.77, *p* = 0.0006, diet: *F*_(1, 44)_ = 8.19, *p* = 0.006; Figure 2L). 

When dividing the 24 h cycle into light/dark periods, differences in open-field activity between strains were more distinct in the light phase. Regardless of diet, Bl6 mice traveled greater distances and spent more time in the center zone on day 1 (Figure 2M,N) and at week 9 (Figure 2S,T) in the light phase. These differences were no longer evident in the dark phase on day 1 (Figure 2P,Q). However, on week 9, Bl6 mice traveled greater distances in the whole arena compared with 129Sv mice (strain: *F*_(1, 44)_ = 39.33, *p* < 0.0001) in the dark phase (Figure 2V), regardless of diet. Furthermore, on week 9, HFD-fed 129Sv mice spent significantly less time in the center zone compared with HFD-fed Bl6 mice (strain: *F*_(1, 44)_ = 9.13, *p* = 0.004; diet: *F*_(1, 44)_ = 5.65, *p* = 0.02) in the dark phase (Figure 2W).

When dividing the 24 h cycle into hourly data, clearly different motor response between Bl6 and 129Sv emerged within the 2 h of behavioral testing (Appendix A). On the first day of the diet, total distance traveled and time spent in the center zone were significantly lower in 129Sv mice compared with Bl6 mice regardless of diet. In addition, 129Sv mice fed HFD spent significantly less time in the center zone than their respective control counterparts during the first hour. At week 9, the total distance traveled remained lower in 129Sv mice compared with Bl6 mice during the first 3 h. However, during the first hour, we observed an interesting HFD-induced motor suppression in 129Sv mice, as HFD-fed mice traveled significantly shorter distances compared with 129Sv mice fed the control diet. At week 9, 129Sv mice also spent significantly less time in the center zone, but this difference was only evident in the first hour. Similar to day 1, 129Sv mice fed with HFD spent significantly less time in the center zone than their corresponding control mice in the first hour.

At week 9, the body weight of Bl6 and 129Sv mice was measured before and after the Phenotyper trial. We observed an interesting strain and diet effect in the animals’ weight (diet: *F*_(1, 43)_ = 13.79, *p* = 0.0006; strain: *F*_(1, 43)_ = 98.78, *p* < 0.0001). Bl6 and 129Sv mice fed with standard diet exhibited weight loss, whereas mice fed with HFD exhibited weight gain. Phenotyper testing induced greater weight loss in 129Sv (−1.10 ± 0.08 g) mice in the control group compared with Bl6 (−0.49 ± 0.10 g) control mice (Figure 2Z). Both strains in HFD groups gained significant weight compared with their control counterparts. However, the weight gain was greater in Bl6 mice in the HFD (1.08 ± 0.23 g) group compared with HFD-fed 129Sv mice (0.50 ± 0.17 g). This could mean that 129Sv mice placed in social isolation conditions could be in a greater state of stress.

Subsequently a Pearson correlation coefficient matrix was created to measure the relationships between the locomotor parameters and 9-week body weight change. In Bl6 mice, 9-week body weight change was positively correlated with total distance traveled in the light phase (*r* = 0.61, *p* = 0.04) and time spent in the center zone in the light phase (*r* = 0.65, *p* = 0.02) (Figure 3A). On the contrary, in 129Sv mice, 9-week body weight change was negatively correlated with total distance traveled in 24 h (*r* = −0.60, *p* = 0.04), in light phase (*r* = −0.66, *p* = 0.02), time spent in the center zone in light phase (*r* = −0.65, *p* = 0.02), and time spent in the food zone in light phase (*r* = −0.59, *p* = 0.04) (Figure 3B).

### 2.3. Metabolic Changes Induced by High-Fat Diet (HFD)

We next examined the metabolic response to a HFD in Bl6 and 129Sv mouse strains. Metabolites from plasma were measured using the AbsoluteIDQ p180 kit (Biocrates Life Sciences AG, Innsbruck, Austria), which detects 186 metabolites in 5 compound classes (acylcarnitines, amino acids, biogenic amines, hexoses, and phospho- and sphingolipids).

#### 2.3.1. HFD Impact on Acylcarnitine Profile

Plasma levels of short-chain acylcarnitines (SCACs) in HFD-fed Bl6 and 129Sv mice compared with the corresponding controls were significantly lower (Figure 4F). More specifically HFD induced a significant decrease in the concentrations of C0, C2, and C4- in Bl6 mice and C0, C2, C3, C4-, and C4:1 in 129Sv mice (Figure 4A,E). Medium-chain acylcarnitines (MCACs) remained unaffected by HFD in both strains. However, plasma levels of long-chain acylcarnitines (LCACs) were specifically altered in HFD-fed 129Sv mice compared with control-diet-fed 129Sv mice (Figure 4L). More precisely, HFD induced a significant decrease in plasma levels of C12, C12:1, C14, C14:2, C16, C16:1, C16:2, C16:2-OH, C18:1-OH, and C18:2 in 129Sv mice. In HFD-fed Bl6 mice, only the concentration of C18:2 was significantly lower compared with control-diet-fed Bl6 mice. C18 was the only AC that exhibited HFD-induced increase in both stains (Figure 4J).

Additionally, the ratio of LCACs to free carnitine [(C16 + C18)/C0] was significantly upregulated by HFD in both strains, which reflects a higher activity of carnitine palmitoyltransferase l (CPT1). This ratio was significantly higher in HFD-fed 129Sv mice compared with HFD-fed Bl6 mice. Additionally, the ratio of the CPT2 [(C16:0 + C18:1)/C2] was higher in HFD-fed Bl6 and 129Sv mice compared with their respective control animals. Moreover, the ratio of dicarboxy-acylcarnitines to total acylcarnitines (total AC − DC/total AC) was significantly elevated in HFD-fed mice of both strains, indicating higher ω-oxidation of fatty acids.

#### 2.3.2. HFD Impact on Amino Acids and Their Derivatives Biogenic Amines

HFD induced an increase in circulating branched-chain amino acids (BCAAs) in 129Sv mice (Figure 5A). While the BCAAs leucine (Leu) and isoleucine (Ile) showed only a trend toward being increased in HFD-fed 129Sv mice, plasma valine (Val) levels (Figure 5B) were significantly increased in HFD-fed 129Sv mice compared with 129Sv control-diet-fed mice. Additionally, HFD resulted in a significant decrease in plasma levels of the non-essential glucogenic amino acid glycine (Gly) in 129Sv mice (Figure 5C). Only the amino acid citrulline (Cit) was affected by HFD in Bl6 mice (Figure 5D). Cit was significantly elevated in HFD-fed Bl6 mice compared with their control counterparts. Furthermore, the ratio of citrulline to ornithine (Cit/Orn) was specifically increased in HFD-fed Bl6 mice (Figure 5E), possibly indicating increased ornithine transcarbamylase activity in Bl6 mice.

HFD induced a significant decrease in kynurenine in both strains (Figure 5F). However, this decrease was significantly more pronounced in Bl6 mice compared with 129Sv. The ratio of kynurenine to tryptophan (Trp) was also significantly reduced after exposure to HFD in both strains, indicating decreased indole dioxygenase activity. Plasma levels of spermidine and trans-4-hydroxyproline (t4-OH-Pro) were significantly diminished in HFD-fed 129Sv compared with control mice, whereas no alterations were observed in Bl6 mice (Figure 5G,H).

#### 2.3.3. HFD Impact on Lipid Metabolism

The total level of lysophosphatidylcholine acyls (LysoPCs; Figure 6A) and sphingomyelins (SMs; Figure 6D) was significantly elevated by HFD in both strains. More precisely, HFD induced an increase in LysoPCs C18:0, C18:1, C18:2, C20:3, C26:0, and C28:1 in both strains. However, LysoPC a C17:0 exhibited HFD-induced decrease in 129Sv mice (Figure 6B). SMs SM(OH) C14:1, SM(OH) C16:1, SM C16:0, SM C16:1, SM C18:0, and SM C18:1 were all significantly elevated in HFD-fed mice of both strains. However, SM (OH) C22:2 exhibited significant HFD-induced decrease in both strains (Figure 6E). Moreover, SM (OH) C22:1, SM C22:3, and SM C24:0 (Figure 6F) were specifically elevated in HFD-fed Bl6 mice compared with control-diet-fed Bl6 mice and no HFD-induced alterations were observed in 129Sv mice. 

Out of 38 PC diacyls (PC aas), 15 were significantly elevated after HFD in both strains (Appendix A). HFD induced an increase in PC aas C30:0, C32:1, C34:2, and C36:4 specifically in Bl6 mice, whereas PC aas C42:2 and C42:5 were specifically decreased in 129Sv mice on HFD. Out of 37 PC acyl-alkyls (PC aes) 26 were significantly affected by HFD (Appendix A). HFD induced a significant elevation in unsaturated (UFA) PC aas and PC aes in both Bl6 and 129Sv mice (Figure 6G,H). When subdividing UFA lipids into polyunsaturated fatty acids (PUFAs) and monounsaturated fatty acids (MUFAs), the HDF-induced increase in 129Sv mice was only evident in MUFA PC aes and did not affect PUFA PC ae levels (Figure 6I,J). However, in Bl6 mice, both PC ae PUFAs and MUFAs were affected by HFD and were both significantly elevated. PUFAs and MUFAs of PC aa species were significantly elevated by HFD in both strains.

### 2.4. Metabolite Differences Highlighted by GLM Analysis

For the association analysis of metabolites, body weight change, and locomotor activity parameters, we used multivariate general linear model (GLM) analysis. To map out the most important differences, we used unpaired *t*-tests and Bonferroni correction to correct for multiple testing (*p* ≤ 0.00027). Significant markers that exceeded the Bonferroni threshold were incorporated into the GLM analysis.

#### 2.4.1. Metabolic Profile Differences between Bl6 and 129Sv

After applying Bonferroni correction (*p* < 0.00027), 45 metabolites remained statistically significant. The vast majority (42) were significantly higher in 129Sv mice, and only 3 were higher in Bl6 mice. These metabolites in Bl6 mice were carnosine, lysoPC a C16:1 and lysoPC a C20:3. Metabolites that were higher in 129Sv included AC C3, BCAAs (Ile, Leu, Val), lysine (Lys), ornithine (Orn), serine (Ser), 6 PC aas, 19 PC aes, and 8 SMs (Appendix A). The following comparison of metabolites in the HFD group altered the list of significantly different markers (Appendix A). Only 26 metabolites remained significantly different after Bonferroni correction and there was a moderate shift toward the Bl6 strain. In Bl6 strain 9 and in 129Sv mice 17 metabolites were significantly higher. In Bl6 mice, carnosine, lysoPC a C16:1 and lysoPC a C20:3 remained higher, as in the case of standard diet. The metabolites included to the list of Bl6 with HFD were alpha-AAA, putrescine, t4-OH-proline, PC aa C32:1, PC aa C34:3, and SM C20:2. The list of 129Sv included 2 PC aas, 9 PC aes, and 6 SMs. The strongest markers favoring 129Sv were sphingolipids SM (OH) C14:1 and SM (OH) C22:2) (*t* values in both cases > 8) (Table 1). This shows a stronger dominance of lipid metabolism in 129Sv mice compared with Bl6 mice. Although not statistically significant after the Bonferroni correction, comparison of hexoses in the HFD groups revealed significantly higher levels in Bl6 mice compared with 129Sv (*t* = 3.03, *p* = 0.006). This is possible showing the stronger impact of glucose metabolism in Bl6 mice.

GLM confirmed a significant main effect (*F*_(1, 16)_ = 908.5; *p* = 0.03) of mouse strain on the levels of several variables in the standard diet group as well as in the HFD group (*F*_(1, 16)_ = 5269.3; *p* = 0.01). Significant biomarkers in the GLM model are highlighted in Table 1. Significant GLM model under basal conditions included total distance traveled in light phase, BCAAs (Ile, Leu, and Val), carnosine, 3 lysoPCs, 6 PC aas, 16 PC aes, and 8 SMs. Significant GLM model under HFD included total distance traveled (light and dark), time spent in center (in light period), 4 biogenic amines (alpha-AAA, carnosine, putrescine, and t4-OH-Pro), 2 lysoPCs, 2 PC aas, 7 PC aes, and 7 SMs. 

In both control- and HFD-fed conditions, the groups showed significantly different metabolite levels between Bl6 and 129Sv, including carnosine, lysoPC a C16:1, 6 PC aes (PC ae C32:1, PC ae C34:2, PC ae C36:2, PC ae C38:2, PC ae C38:6 and PC ae C40:6), 6 SMs (SM (OH) C14:1, SM (OH) C16:1, SM (OH) C22:1, SM (OH) C22:2, SM C16:0, and SM C16:1) and total distance traveled in light period.

#### 2.4.2. GLM Analysis of HFD-Induced Alterations in Metabolic Profile of Bl6 Mice

GLM confirmed a significant main effect (*F*_(1, 20)_ = 360.13, *p* = 0.04) of diet on metabolite levels in Bl6 mice. The final GLM model retained 47 metabolites: 5 ACs (C0, C2, C4-, C18 and C18:2), 1 biogenic amine (kynurenine), 4 lysoPCs, 13 PC aas, 16 PC aes, and 8 SMs (Table 2). The majority of metabolites were increased due to HFD. The exceptions were ACs C0, C2, C4-, C18:2, and kynurenine, being decreased with HFD. The strongest associations (ß > 0.90) in Bl6 were established for C0, 5 PC aas, 4 aes, and SM C16:0.

#### 2.4.3. GLM Analysis of HFD-Induced Alterations in Metabolic Profile of 129Sv Mice

GLM confirmed a significant main effect (*F*_(1, 16)_ = 811.11, *p* = 0.03) of diet in 129Sv mice. Similar to Bl6 mice, most HFD-induced alterations were observed in the lipid profile. The final GLM model retained 9-week weight gain and 48 metabolites: 6 ACs (C0, C2, C3, C4-, C14.2, C18:2), 1 amino acid (Gly), 2 biogenic amines (kynurenine and t4-OH-Pro), 5 lysoPCs, 13 PC aas, 16 PC aes, and 6 SMs (Table 2). The majority of metabolites were increased due to HFD. The exceptions were ACs C0, C2, C3, C4-, C14:2, C18:2, amino acids and their derivatives Gly, t4-OH-Pro, kynurenine and SM (OH) C22:2, being decreased with HFD. The strongest associations (ß > 0.90) in 129Sv were established for C0, lysoPC a C20:3, 5 PC aas, PC ae C38:2, and sphingolipids (SM (OH) C14:1, SM C16:0). It is worthy to note that associations of C0, PC aa 36:1, PC aa C36:2, PC aa C36:3, PC aa C38:3, and SM C16:0 were overlapping in Bl6 and 129Sv.

## 3. Discussion

Metabolomics and metabolic profiling have become powerful tools for investigating metabolic processes, identifying potential biomarkers, and characterizing various pathological conditions. Previous studies have demonstrated that different inbred mouse strains vary substantially in their metabolic phenotype under physiological and pathological conditions. In previous research, we have demonstrated that there is significant metabolic heterogeneity between two commonly used mouse strains: C57BL/6NTac (Bl6) and 129S6/SvEvTac (129Sv). We have repeatedly shown that metabolites C4-, C5-, and SM(OH) C22:2 are significantly higher in 129Sv mice and carnosine, alpha-AAA, Ac-Orn, and lysoPC a C16:1 belong to the metabolic signature of Bl6 mice [2,5]. Moreover, we have recently shown that systemic administration of lipopolysaccharide (LPS) leads to hypometabolism (which is beneficial for host tolerance) in the Bl6 strain but increases the production of proinflammatory metabolites in the 129Sv strain [5]. Several studies have also indicated that consumption of high-fat foods may induce systemic inflammation [16,17,18]. Thus, in this study, we aimed to investigate similarities and differences in a number of metabolic variables and their ratios in Bl6 and 129Sv mouse strains in response to a 9-week high-fat diet (HFD). 

### 3.1. HFD Causes Weight Gain in 129Sv Mice, but Not in Bl6 Mice

Bl6 and 129Sv mice were fed HFD or regular chow diets for 9 weeks. Body weight, food, and water intake were recorded weekly. HFD-fed 129Sv mice began to weigh significantly more than mice fed with standard chow starting from the second week of dietary exposure and continuing to week 9. On the contrary, body weight gain of Bl6 mice was not affected by HFD during the 9-week study period. GLM analysis confirmed significant 9-week body weight gain of 129Sv mice. Interestingly, the weight gain of 129Sv mice was not related to higher amounts of food consumed as the food intake was significantly lower in both 129Sv and Bl6 HFD groups compared with their respective control-diet groups. The difference in food consumption between HFD and control groups became evident starting from the second week of dietary exposure and remained lower until the end of dietary exposure. A similar effect on food consumption has previously been observed in our laboratory [19]. Taken together, while 129Sv mice responded strongly with weight gain to HFD, the Bl6 strain was protected against HFD-induced increase in body weight.

### 3.2. HFD Reduces Locomotor Activity in 129Sv Mice, While Bl6 Mice Visit Less HFD Food Zone

Locomotor activity of Bl6 mice in the open-field test remained the same regardless of diet, whereas HFD-exposed 129Sv mice traveled significantly shorter distances compared with their control diet counterparts, as well as compared with HFD-fed Bl6 mice already at the beginning of the dietary exposure (day 1) and at the end of the dietary exposure (week 9). HFD also induced higher anxiety-like behavior in 129Sv mice, as they spent significantly less time in the center zone of the open field at week 9 than control-diet-fed 129Sv mice. When the 24 h cycle was divided into light and dark phases, the difference between strains became even more prominent in the light phase. However, at week 9, HFD-fed 129Sv mice tended to spend less time in the center zone, regardless of the light or dark phase of the 24 h cycle. On the other hand, HFD-exposed Bl6 mice visited the food zone significantly less at both the beginning and end of the dietary exposure. Additionally, when splitting 24 h cycle into hourly data, clearly different motor response between Bl6 and 129Sv emerged within 2 h from the beginning of behavioral testing on the first day. 129Sv mice traveled significantly shorter distances in the total arena and spent significantly less time in the center zone than Bl6 mice during the first two hours. This difference was no longer evident from the third hour onward. This observation most likely reflects a higher anxiety-like trait of 129Sv at the beginning of behavioral testing, indicating passive adaptation. At week 9, the total distance traveled remained significantly different between strains even at the third hour. On the other hand, 129Sv spent significantly less time in the center zone only at the first hour and, starting from the second hour, it was already equivalent to that of Bl6 mice. In addition, we observed that at the end of the dietary intervention, the HFD-fed 129Sv mice traveled significantly shorter distances and spent significantly less time in the center zone than the control-diet-fed 129Sv mice in the first hour. This suggests that HFD exacerbates the anxiety-like state in 129Sv mice. Psychiatric disorders and type II diabetes mellitus (T2DM) have been shown to be highly comorbid, and this may suggest that the 129Sv strain is well-suited for modeling the metabolic syndrome associated with psychiatric disorders.

Additionally, we found that body weight gain of 129Sv mice was negatively correlated with total distance traveled and duration in the center and food zones in the light phase. By contrast, weight gain of Bl6 mice was positively correlated with total distance traveled in the light phase and time spent in the center zone in the light phase. Thus, the increased body weight of 129Sv mice caused a decrease in the activity, or *vice versa*. After the second open-field test in Phenotyper cages, we observed an interesting effect on weight in both strains. Control mice of both strains lost significant weight during the exposure; however, this weight loss was greater in 129Sv mice. In animals exposed to HFD, an opposite effect emerged, as mice of both strains exhibited weight gain during the exposure. However, it was more pronounced in Bl6 mice than in 129Sv mice, which differed from the general dynamics of body weight in the home cage. This effect cannot be explained by increased locomotor activity of 129Sv mice, since it was significantly lower compared with Bl6 mice. Thus, it appears that under stressful conditions, such as social isolation, 129Sv mice exhibit greater weight loss, which can be prevented by HFD. Indeed, 129Sv mice have been shown to gain more body weight in a non-stressful home-cage environment compared with Bl6 mice. However, after the repeated stressful interventions, 129Sv mice tend to lose body weight not seen in Bl6 mice [2]. It is common knowledge that Bl6 mice are more capable of withstanding stress and adapting to new environments than 129Sv mice [3,4,20], and decreased activity and greater weight loss of 129Sv reflects inability to cope in a novel environment.

### 3.3. Metabolic Profile Differences between Bl6 and 129Sv

Metabolic profiling of Bl6 and 129Sv blood samples revealed highly distinct variations between strains at baseline level. In total, we identified 78 metabolites that differed significantly between the two groups. Among these metabolites, concentration differences of C4-, C5-, carnosine, alpha-aminoadipic acid (alpha-AAA), sphingolipid SM(OH) C22:2, and lysophosphatidylcholine acyl (lysoPC a) C16:1 have also been shown to differ between these strains in previous studies [2,5]. Higher levels of C4-, C5-, and SM(OH) C22:2 have been shown to be part of the metabolic signature of 129Sv. C4- is a mixture of two isomers: butyrylcarnitine, derived from fatty acid metabolism; isobutyryl carnitine, derived from Val metabolism [21]. Accordingly, the amino acid Val was also significantly higher in 129Sv mice. However, in contrast to previous studies, we did not find equivalent differences in AC C5- levels when comparing Bl6 and 129Sv mouse lines. The nature of this difference requires further research. On the other hand, the higher concentrations of carnosine, alpha-AAA, and lysoPC a C16:1 belong to the metabolic signature of Bl6. Carnosine is an endogenous dipeptide distributed widely in skeletal muscles, heart, and the central nervous system (CNS) [22]. Carnosine has been described to have antioxidant properties and has been shown to scavenge reactive oxygen species (ROS) [23,24]. In addition, carnosine has been found to act as a scavenger of reactive aldehydes from the oxidative degradation pathway of endogenous molecules such as sugars, polyunsaturated fatty acids (PUFAs), and proteins [25]. Furthermore, carnosine supplementation has been shown to reduce lipid accumulation in the circulation and attenuate HFD-induced hepatic steatosis [26]. Thus, higher carnosine levels could be beneficial and contribute to protection against HFD-induced disorders in Bl6 mice. Higher levels of alpha-AAA in Bl6 mice are reportedly caused by a defect in the *Dhtkd1* gene, which has been identified as a primary regulator of alpha-AAA, and defects in this gene lead to accumulation of alpha-AAA [6,13]. Alpha-AAA is an intermediate in the Lys metabolic pathway, a marker of oxidative stress and a biomarker of insulin resistance and diabetes risk [7,8,9]. On the other hand, recent work has demonstrated that treatment of diet-induced obesity in mice with alpha-AAA reduces body weight and decreases fat accumulation [27]. 

GLM analysis confirmed significant strain differences in Ile, Leu, Val, and carnosine levels. In addition, significant differences between strains were observed in lipid profile. We identified 3 lysoPCs, 6 PC aas, 16 PC aes, and 8 SMs to be significantly different between Bl6 and 129Sv mice. After the exposure to HFD metabolite differences between strains included alpha-AAA, carnosine, putrescine, t4-OH-Pro, 2 lysoPCs, 2 PC aas, 7 PC aes, and 7 SMs. This shows that HFD somewhat affects the differences between strains; nevertheless, the most significant differences remain (e.g., carnosine and lysoPC a C16:1).

### 3.4. HFD Induces a Greater Number of Alterations in Acylcarnitine Profile of 129Sv Mice

Nine weeks of HFD induced a wide range of alterations in several different metabolite groups. Many of the metabolic shifts caused by HFD were similar in Bl6 and 129Sv mouse strains. HFD induced a decrease in carnitine (C0) and short-chain acylcarnitines (SCACs) (C2, C4-) in both mouse strains. However, the altered profile of SCACs was wider in 129Sv mice and additionally included C3 and C4:1. AC C3 is a by-product of Ile and Val catabolism, which were elevated specifically in 129Sv mice [28]. Carnitine is crucial in the degradation of long-chain fatty acids in the mitochondria and specifically important for making energy from food enriched with fats. Plasma levels of long-chain acylcarnitines (LCACs) were especially altered by HFD in 129Sv mice. More precisely, HFD induced a significant decrease in plasma levels of C12, C12:1, C14, C14:2, C16, C16:1, C16:2, C16:2-OH, C18:1-OH, and C18:2 in 129Sv mice. Only the concentration of C18:2 was decreased in HFD-fed Bl6 mice. C18 was the only AC that exhibited HFD-induced increase in both stains. Given that C18 may produce a significant amount of ATP, the increase is likely a result of compensating for the energetic demand. LCACs are intermediates of intracellular fatty acid metabolism that are generated by transesterification of long-chain acyl-CoA with carnitine-by-carnitine palmitoyltransferase l (CPT1). In turn, the elevation of C18 caused upregulation in the ratios of long-chain species to free carnitine and ACs in both HFD-fed strains, reflecting increased activity of CPT1 and CPT2. Long-chain fatty acids are transported into mitochondria via CPT1 and CPT2, which are located in the outer and inner mitochondrial membrane, respectively, and oxidized via the β-oxidation pathway for energy production [29]. The fact that both strains had increased CPT1 and CPT2 activity indicates higher uptake of fatty acids into mitochondria and possibly increased oxidation rate of long-chain fatty acids. Upregulation of β-oxidation is expected as the HFD is responsible for the overload of fatty acid metabolism. The decreased C0 levels could suggest insufficient β-oxidation to compensate for the potential HFD-induced elevation of free fatty acids. In addition, we observed an increase in the ratio of dicarboxy-acylcarnitines to total ACs in HFD-fed mice of both strains, indicating intensification of the ω-oxidation pathway. Activation of ω-oxidation has been described as a rescue mechanism for fatty acid disorders, as it could potentially alleviate the overload of lipid catabolism pathways [30]. GLM model confirmed significant HFD-induced decrease in ACs C0, C2, C4-, and C18:2 in both strains. ACs C3 and C14:2 were specifically decreased in 129Sv mice and C18 was specifically increased in Bl6 mice. The classical understanding is that ACs are transported into mitochondria for the purpose of energy production; however, under HFD conditions, they are also used for triglyceride synthesis. Thus, our results might suggest that 129Sv mice cannot adapt as effectively as Bl6 mice and that ACs may also be used in the production of lipid droplets in muscle cells. This could also explain why 129Sv mice gain more weight and become less active compared with Bl6 mice.

### 3.5. HFD Induces More Changes in Amino Acids and Their Derivatives Biogenic Amines in 129Sv Mice

HFD induced an increase in circulating ketogenic and branched-chain amino acids (BCAAs) in 129Sv mice, whereas no changes in BCAA levels were observed in Bl6 mice. Although the ketogenic BCAAs Leu and Ile were slightly increased in HFD-fed 129Sv mice, the level of Val, another BCAA, was significantly elevated by HFD, suggesting disturbances in Val metabolism. BCAAs can be used for the fast production of ketone bodies. Furthermore, as Leu is a direct trigger for protein synthesis, it may further suggest the production and accumulation of lipid droplets in muscle cells, as these amino acids will be used for protein synthesis and the formation of ketone bodies. Increases in BCAAs have also been shown to be associated with obesity and in HFD-fed animals BCAA contributes to development of obesity-associated insulin resistance [28]. The same study also reported that obese subjects have significantly lower Gly levels. A recent study demonstrated that BCAA restriction could prevent excessive weight gain, adipose tissue accumulation, and adipocyte hypertrophy induced by HFD. In addition, BCAA restriction in HFD helped maintain normal glucose and insulin levels and prevented insulin resistance [31]. In addition, low plasma Gly concentrations have been reported to be associated with T2DM [32]. Correspondingly, we observed that HFD induced a significant decrease in Gly in 129Sv mice. In Bl6 mice, the only amino acid affected by HFD was Cit, which was significantly increased. Orn levels were slightly lower in HFD-fed Bl6 mice, although this result was not statistically significant. Consequently, we observed an increased ratio of Cit to Orn in HFD-fed Bl6 mice, possibly indicating increased ornithine transcarbamylase activity, suggesting disturbances in urea cycle in Bl6 mice. 

HFD affected the kynurenine pathway in both mouse strains. Mice exhibited HFD-induced decrease in kynurenine and in the ratio of kynurenine to Trp, indicating decreased indole dioxygenase activity and diminished Trp breakdown. Kynurenine and Trp have been reported to be positively associated with obesity [33]. Trp is metabolized via the kynurenine pathway, producing kynurenine, which is further metabolized in three distinct routes to quinolinate, kynurenate, and xanthurenate, producing glutamate from α-ketoglutarate, which in turn is a key molecule in the TCA cycle. A small percentage of Trp is also hydroxylated to synthesize serotonin and melatonin. In our study, the fact that kynurenine levels were elevated in the HFD groups suggests an upregulation of the kynurenine pathway that could lead to an overproduction of xanthurenic acid, considered as one of factors promoting insulin resistance [34]. In addition, 129Sv mice had decreased plasma levels of spermidine and t4-OH-Pro when consuming HFD. T4-OH-Pro is a major component of the protein collagen. It is produced by hydroxylation of the amino acid proline (Pro) and is, therefore, a post-translationally modified, non-essential amino acid. T4-OH-Pro and Pro play a key role in collagen stability, and the decreased concentrations seen here may reflect increased turnover of collagen. GLM analysis confirmed significant HFD-induced alterations of kynurenine in both strains and revealed t4-OH-Pro to belong specifically in 129Sv metabolic signature of HFD.

### 3.6. HFD Induces More Changes in Lipid Profile of Bl6 Mice

Most altered metabolites in plasma were found to be intermediates of lipid metabolism. Lipids are an important source and store of energy for metabolism. Both strains exhibited HFD-induced increase in the total level of lysoPCs, SMs, PC aas, and PC aes. Out of 13 lysoPCs, HFD induced the elevation of 6 in both strains. We identified two additional lysoPCs that were especially altered in HFD-fed 129Sv mice. HFD significantly increased lysoPC a C20:4 in 129Sv mice and in contrast to the general pattern of lysoPCs and lysoPC a C17:0 was significantly decreased. LysoPCs are bioactive proinflammatory lipids, which are hydrolyzed derivatives of PCs and have been shown to have a role in development of atherosclerosis and hyperlipidemia [35,36]. Elevated lysoPC levels might indicate HFD specific inflammatory response. HFD-induced alterations in SMs in both strains included increased levels of six SMs and significantly decreased levels of SM (OH) C22:2. However, we identified additional three SMs that were especially elevated in Bl6 mice—SM (OH) C22:1, SM C22:3, and SM C24:0. We found that the concentrations of PCs were also significantly altered. A total of 15 PC aas and 26 PC aes were significantly elevated in both strains. The level of saturated fatty acids (SFA) of PC aa and ae species were not affected by HFD. However, we observed a significant HFD-induced elevation in unsaturated fatty acid (UFA) PC aa and ae species in both Bl6 and 129Sv mice. When UFA lipids were further subdivided into monounsaturated fatty acids (MUFAs) and polyunsaturated fatty acids (PUFAs), the HFD-induced increase in both strains remained significant only for PC aa species. In 129Sv mice, HFD did not affect PUFA PC ae levels, and the HFD-induced increase was only evident in MUFA PC ae species. On the other hand, both PUFAs and MUFAs of PC ae species were significantly elevated in Bl6 mice. HFD-induced changes in individual PC aas and PC aes were significantly wider in Bl6 mice than 129Sv.

Considering that lysoPCs are metabolites of PC aas, it is important to highlight the following relations under the influence of HFD in Bl6 mice. Indeed, there is a clear relation in Bl6 mice between lysoPC a C16:1 and PC aa C32:1 as well as between lysoPC a C20:3 and PC aa C34:3. Therefore, one can suggest that processing of certain lipids is intensified under the influence of HFD in Bl6 mice.

GLM analysis confirmed that HFD induced a wide range of alterations in lipid profiles. Altogether, 30 lipids were significantly altered in both Bl6 and 129Sv mice. Specifically, in Bl6 mice, PC aa C32:1, PC aa C34:2, PC aa C38:0, PC aa C38:4, PC ae C36:2, PC ae C36:4, PC ae C38:4, PC ae C40:5, SM C18:0, SM C22:3, and SM C24:0 were significantly elevated by HFD. In 129Sv mice, lysoPC a C28:1, PC aa C32:3, PC aa C38:1, PC aa C40:5, PC ae C36:5, PC ae C38:2, PC ae C40:4, and PC ae C42:0 were significantly elevated, while SM (OH) C22:2 was significantly decreased by HFD.

## 4. Materials and Methods

### 4.1. Mouse Strains and Grouping

Wild-type male mice (4-month-old) from the two inbred strains, C57BL/6NTac (Bl6, n = 28) and 129S6/SvEvTac (129Sv; n = 30), were housed in the Laboratory Animal Center at University of Tartu. Mice from both strains were randomly assigned to the standard chow control group (Bl6, n = 13; 129Sv, n = 15) or high-fat diet (HFD; Bl6, n = 14; 129Sv, n = 15) dietary intervention group (Figure 7). Mice were kept under standard conditions with unlimited access to food and water on a 12/12 h light/dark cycle. All animal procedures were conducted in accordance with the European Communities Directive (2010/63/EU) with permit (No. 141, 17 April 2019) from the Estonian National Board of Animal Experiments.

### 4.2. Composition of Diets

Mice were fed either standard chow diet (V1534-300 rat/mice universal maintenance diet; Ssniff Spezialiäten GmbH, Soest, Germany) or high-fat diet (HFD) chow (D12451, 45 kJ% fat (lard); Ssniff Spezialiäten GmbH, Soest, Germany) for 9 weeks. Diet compositions are shown in Table 3.

### 4.3. Measurement of Body Weight and Food and Water Intake

All mice were weighed weekly starting from day one of dietary intervention and the body weight was tracked for 9 weeks during the whole study period.

Food and water intake were recorded weekly during the 9-week study period. Food and water consumption was measured by subtracting the mass (g) or volume (mL) of remaining food and water from the initial food and water mass or volume provided to each cage weekly. For analysis, food and water consumption data were corrected for the number of animals per cage.

### 4.4. Open-field test

Open-field testing was conducted at week 1 (day one of dietary intervention) and week 9 (at the end of dietary intervention). Mice were monitored for 24 h in PhenoTyper^®^ (EthoVision 3.0, Noldus Information Technology, Wageningen, The Netherlands) cages. In this 24 h period, mice were individually housed in 30 cm × 30 cm × 35 cm plexiglass cages with sawdust bedding. Mice had free access to food and water throughout the testing period. Mice were kept under a 12:12 h light/dark cycle. Each cage was equipped with a top unit with integrated infrared sensitive camera and infrared LED lights, which makes tracking possible in the dark phase. The open-field arena was virtually divided into central, peripheral, and food and water zones. The center zone was defined as half of the overall area of the test arena. The total distance traveled, time spent in the center and the food and water zone were measured. Animal movements were continuously recorded by a video-tracking system.

### 4.5. Sample Collection

Mice were euthanized by decapitation, and trunk blood was collected into EDTA-coated microcentrifuge tubes and stored on ice. All the tubes were centrifuged at 2000× *g* for 15 min at 4 °C. Plasma supernatant was separated and stored at −80 °C until further analysis.

### 4.6. Measurement of Metabolites

Targeted metabolomics test kit AbsoluteIDQ p180 (BIOCRATES Life Sciences AG, Innsbruck, Austria) was used to measure plasma levels of 186 different compounds. Samples were determined using mass-spectrometry QTRAP 4500 (Sciex, Framingham, MA, USA) and high-performance liquid chromatography (HPLC) (Agilent 1260 series, Agilent Technologies, Waldbronn, Germany). The first step of sample preparation was performed on the AbsoluteIDQ kit plate included in the test kit with 96 wells to hold the zero-sample, three samples of phosphate-buffered saline, seven calibration standards, and at least three quality controls. Plasma samples were thawed and centrifuged at 4 °C for 5 min at 2750× *g*. To all wells, except the well in position A1, 10 mL of the internal standard mixture was added. Then, 10 mL of the calibration standards, phosphate-buffered saline, quality controls, and plasma samples were added to the respective wells. To each well, 50 μL of a 5% solution of phenyl isothiocyanate in pyridine/ethanol/water (1:1:1, *v*/*v*/*v*) was added for amino acid derivatization. After 20 min of incubation, the plate was dried at room temperature under dry air flow, and all compounds were extracted into solution with 300 μL 5 mM ammonium acetate in methanol. After shaking for 30 min, the plate was centrifuged and the extract filtered through a filter membrane into the 96-well capture plate below. From the capture plate, 50 μL of the solution was transferred to another 96-well plate and diluted with 250 μL of 40% (*v*/*v*) methanol in water for liquid chromatography techniques (LC-MS). For flow injection analysis (FIA), 20 μL of the solution was transferred to another 96-well plate and diluted using 380 μL of FIA mobile phase solvent which was prepared by diluting Biocrates Solvent I provided with the kit in 290 mL of HPLC methanol. Amino acids and biogenic amines in the samples were measured using the LC-MS techniques. Acylcarnitines (Cx:y), hexoses, sphingolipids [SMx:y or SM (OH)x:y], glycerophospholipids (lysophosphatidylcholines (lysoPCx:y), and phosphatidylcholines (PCaa x:y and PC ae x:y)) were measured by FIA tandem mass spectrometry. Multiple reaction monitoring was used for both analytical methods. Concentrations of metabolites were automatically calculated in mM using MetIDQTM software (BIOCRATES Life Sciences AG).

### 4.7. Statistical Analysis

Results are expressed as mean values ± SEM. Statistical analyses were performed using GraphPad Prism 9 software (GraphPad, San Diego, CA, USA) and TIBCO StatisticaTM Version 13.3.0. The Shapiro–Wilk test was applied to assess the normality of data distribution. To normalize the distribution of locomotor and metabolomic data, logarithmic transformation (log2) of the values was performed prior to data analysis. Statistical analysis was performed using two-way ANOVA (strain × diet) or repeated-measures ANOVA (time × diet) followed by Bonferroni post hoc test. Associations between body weight gain and locomotor parameters were analyzed using the Pearson correlation. General linear model (GLM) analysis with backward elimination procedure was performed to examine the associations between body weight change, locomotor parameters, and metabolites. All differences were considered statistically significant at *p* ≤ 0.05.

## 5. Conclusions

The present study expands our understanding of the stress-coping properties of two mouse lines, Bl6 and 129Sv, confirming that 129Sv mice are better suited to study anxiety-, depressive-, and psychosis-like conditions. At the same time, the Bl6 line is favored in the study of social dominance, aggression, addictive behavior, and conditions requiring rapid adaptation. In the case of psychiatric diseases, such as schizophrenia spectrum disorders, major depressive disorder, and bipolar disorder, the development of obesity and metabolic syndrome occurs in patients parallel to the development of the disease, which reduces the effectiveness of treatment and complicates the further course of the disease. In the present work, we found that, under established conditions, the 129Sv mouse line exhibits significant weight gain on HFD with similar metabolite shifts to those seen in humans with metabolic syndrome. In addition, we found that 129Sv mice tend to gain significantly more weight on a normal diet [2]. Thus, the results of our studies confirm that the 129Sv mouse line can be a good model for modeling the metabolic syndrome associated with psychiatric disorders.

## Figures and Tables

**Figure 1 ijms-23-11682-f001:**
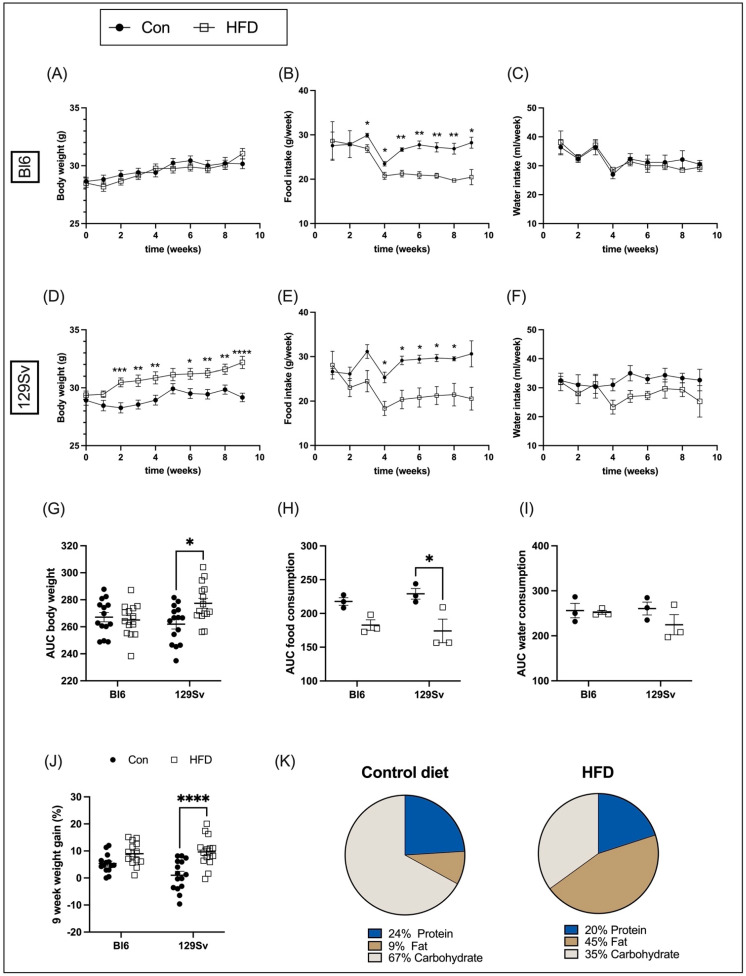
Effects of HFD on body weight and food and water intake in Bl6 and 129Sv mice. Weekly body weight measurements of Bl6 (**A**) and 129Sv (**D**) mice during 9 weeks of HFD experiment. Weekly measurements of food (**B**,**E**) and water (**C**,**F**) intake of Bl6 (**B**,**C**) and 129Sv (**E**,**F**) mice. AUC of body weight (**G**), food (**H**), and water (**I**) intake. (**J**) Nine-week weight gain. (**K**) Macronutrient content of control diet and HFD. Data are expressed as mean values ± SEM. * *p* ≤ 0.05, ** *p* ≤ 0.01, *** *p* ≤ 0.001, **** *p* ≤ 0.0001 (diet effect).

**Figure 2 ijms-23-11682-f002:**
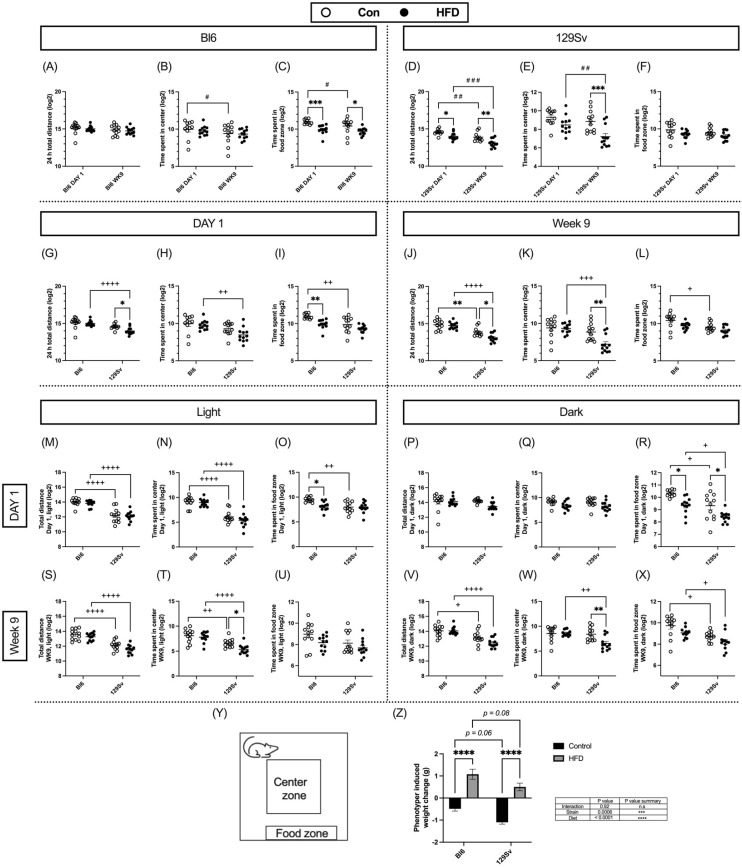
HFD impact on locomotor parameters (Log2 values, data expressed as mean ± SEM). Bl6: Total distance traveled in 24 h (**A**), time spent in center (**B**) and food (C) zone day 1 vs. week 9. 129Sv: Total distance traveled in 24 h (**D**), time spent in center (**E**) and food (**F**) zone day 1 vs. week 9. At the beginning of dietary intervention: Total distance traveled in 24 h (**G**), time spent in center (**H**) and food (**I**) zone Bl6 vs. 129Sv. Week 9 of dietary intervention: Total distance traveled in 24 h (**J**), time spent in center (**K**) and food (**L**) zone Bl6 vs. 129Sv. Total distance traveled at day 1 (**M**) and week 9 (**S**) at lights on period. Total distance traveled on day 1 (**P**) and week 9 (**V**) at dark period. Time spent at the center zone on day 1 (**N**) and week 9 (**T**) at lights on period. Time spent at the center zone on day 1 (**Q**) and week 9 (**W**) at dark period. Time spent at the food zone on day 1 (**O**) and week 9 (**U**) at lights on period. Time spent at the food zone on day 1 (**R**) and week 9 (**X**) at dark period. (**Y**) Representation of Phenotyper test arena indicating the location of defined zones of interest including food and water zone and center zone. (**Z**) Weight change (g) of Bl6 and 129Sv mice induced by week 9 Phenotyper testing. * *p* ≤ 0.05, ** *p* ≤ 0.01, *** *p* ≤ 0.001, **** *p* ≤ 0.0001 (diet effect), + *p* ≤ 0.05, ++ *p* ≤ 0.01, +++ *p* ≤ 0.001, ++++ *p* ≤ 0.0001 (strain effect). # *p* ≤ 0.05, ## *p* ≤ 0.01, ### *p* ≤ 0.001 (time effect).

**Figure 3 ijms-23-11682-f003:**
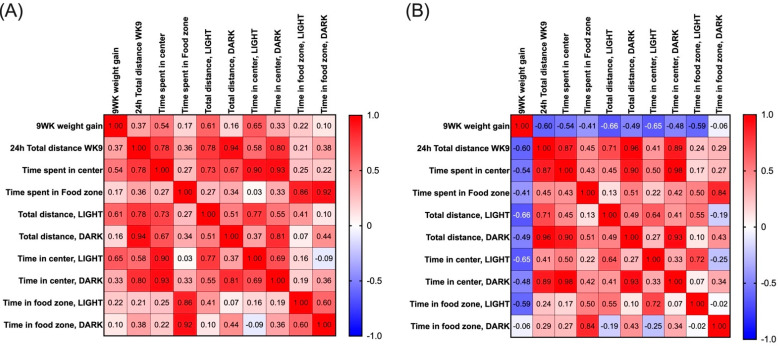
Heat-map of Pearson correlation coefficients between 9-week weight gain and locomotor parameters in (**A**) Bl6 and (**B**) 129Sv HFD-fed mice.

**Figure 4 ijms-23-11682-f004:**
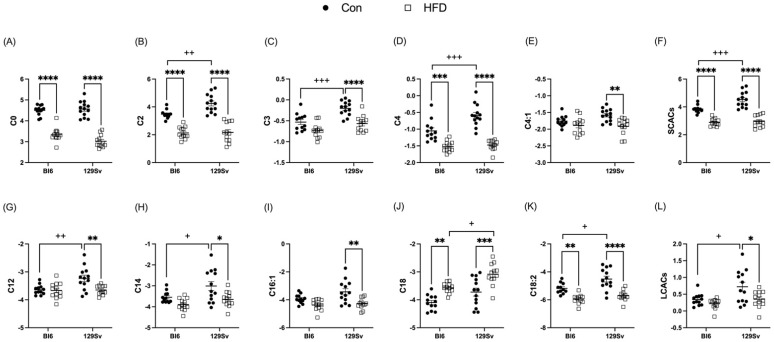
Effect of HFD on the level of selected acylcarnitines (Log2 values, data expressed as mean ± SEM). The level of short-chain acylcarnitines (**A**) C0, (**B**) C2, (**C**) C3, (**D**) C4-, and (**E**) C4:1 and (**F**) the sum of SCACs. The level of long-chain acylcarnitines (**G**) C12, (**H**) C14, (**I**) C16:1, (**J**) C18, and (**K**) C18:2 and (**L**) the sum of LCACs. Two-way ANOVA (Bonferroni post hoc test): * *p* ≤ 0.05, ** *p* ≤ 0.01, *** *p* ≤ 0.001, **** *p* ≤ 0.0001 (diet effect), + *p* ≤ 0.05, ++ *p* ≤ 0.01, +++ *p* ≤ 0.001 (strain effect).

**Figure 5 ijms-23-11682-f005:**
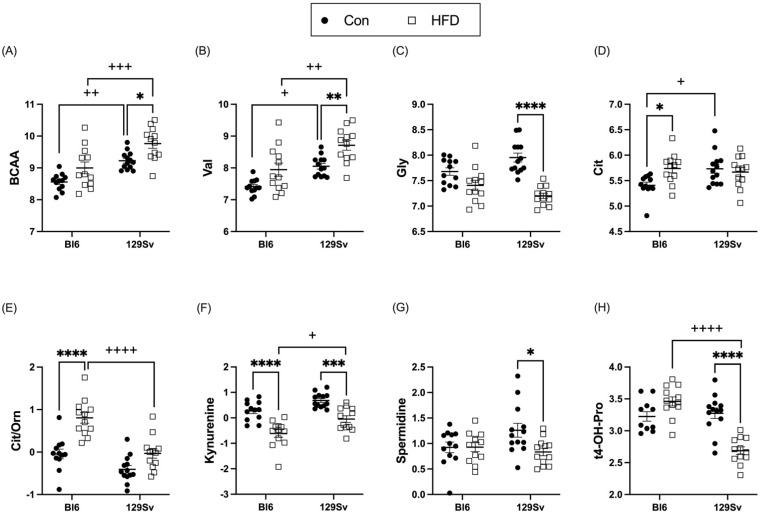
Effect of HFD on the level of selected amino acids and their derivatives biogenic amines (Log2 values, data expressed as mean ± SEM). (**A**) sum of branched-chain amino acids (BCAAs), (**B**) valine (Val), (**C**) glycine (Gly), (**D**) citrulline (Cit), (**E**) ratio of citrulline to ornithine (Cit/Orn), (**F**) kynurenine, (**G**) spermidine, and (**H**) trans-4-hydroxyproline (t4-OH-Pro). Two-way ANOVA (Bonferroni post hoc test): * *p* ≤ 0.05, ** *p* ≤ 0.01, *** *p* ≤ 0.001, **** *p* ≤ 0.0001 (diet effect), + *p* ≤ 0.05, ++ *p* ≤ 0.01, +++ *p* ≤ 0.001, ++++ *p* ≤ 0.0001 (strain effect).

**Figure 6 ijms-23-11682-f006:**
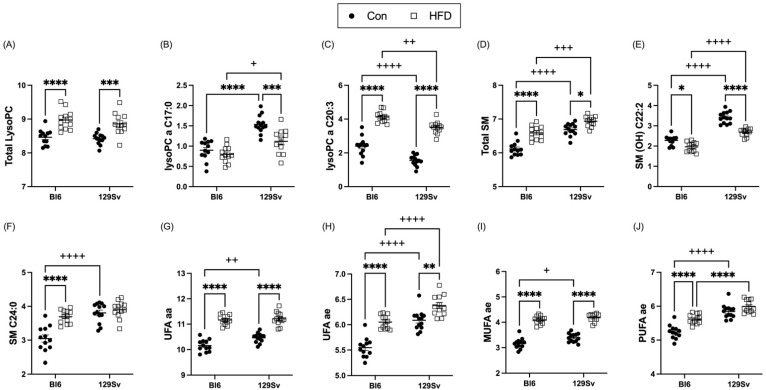
Effect of HFD on the level of selected lipids (Log2 values, data expressed as mean ± SEM). (**A**) sum of lysoPCs, (**B**) lysoPC a C17:0, (**C**) lysoPC a C20:3, (**D**) sum of sphingomyelins, (**E**) SM (OH) C22:2, (**F**) SM C24:0, unsaturated (UFA) PC aas, (**G**) and PC aes (**H**), (**I**) monounsaturated fatty acids (MUFA) and (**J**) polyunsaturated fatty acids (PUFA) of PC ae species. Two-way ANOVA (Bonferroni post hoc test): * *p* ≤ 0.05, ** *p* ≤ 0.01, *** *p* ≤ 0.001, **** *p* ≤ 0.0001 (diet effect), + *p* ≤ 0.05, ++ *p* ≤ 0.01, +++ *p* ≤ 0.001, ++++ *p* ≤ 0.0001 (strain effect).

**Figure 7 ijms-23-11682-f007:**
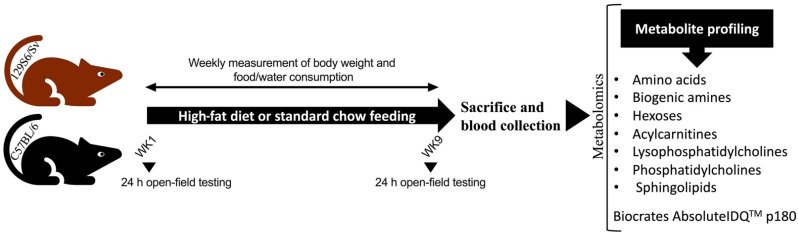
Schematic overview of experimental design. Bl6 and 129Sv mice were fed with a high-fat diet or standard chow for 9 weeks. Body weight and food and water consumption were measured weekly. Locomotor activity was assessed at the start of the dietary intervention and week 9. Subsequently, mice were sacrificed, and blood was collected for metabolomic analysis.

**Table 1 ijms-23-11682-t001:** Effect of mouse strain on blood plasma levels of metabolites and locomotor parameters between standard diet groups and HFD groups.

Standard Diet				HFD			
	ß (95% CI)	*t*-Value	*p*-Value		ß (95% Cl)	*t*-Value	*p*-Value
Total distance; light	0.76 (0.46, 1.05)	5.29	0.00003	Total distance	0.81 (0.48, 1.13)	5.26	0.0001
Ile	−0.77 (−1.06, −0.49)	−5.60	0.00002	Total distance; light	0.83 (0.52, 1.14)	5.67	0.00005
Leu	−0.76 (−1.06, −0.47)	−5.37	0.00003	Center time	0.72 (0.34, 1.10)	4.06	0.001
Val	−0.75 (−1.05, −0.44)	−5.14	0.00004	Center time; light	0.83 (0.52, 1.14)	5.78	0.00004
Carnosine	0.87 (0.64, 1.09)	7.92	<0.00001	alpha-AAA	0.85 (0.56, 1.14)	6.24	0.00002
lysoPC a C16:1	0.81 (0.55, 1.08)	6.42	<0.00001	Carnosine	0.81 (0.49, 1.13)	5.52	0.00007
lysoPC a C17:0	−0.83 (−1.08, −0.57)	−6.72	<0.00001	Putrescine	0.71 (0.32, 1.10)	3.88	0.001
lysoPC a C24:0	−0.77 (−1.06, −0.48)	−5.58	0.00002	t4-OH-Pro	0.79 (0.46, 1.13)	5.07	0.0001
PC aa C30:2	−0.77 (−1.06, −0.48)	−5.54	0.00002	lysoPC a C16:1	0.71 (0.32, 1.10)	3.89	0.001
PC aa C36:0	−0.73 (−1.04, −0.42)	−4.94	0.00007	lysoPC a C20:3	0.71 (0.32, 1.10)	3.86	0.002
PC aa C36:2	−0.85 (−1.09, −0.60)	−7.26	<0.00001	PC aa C32:1	0.70 (0.31, 1.10)	3.85	0.002
PC aa C40:2	−0.74 (−1.04, −0.43)	−4.98	0.00006	PC aa C34:3	0.81 (0.49, 1.13)	5.40	0.00007
PC aa C40:5	−0.77 (−1.06, −0.48)	−5.47	0.00002	PC ae C32:1	−0.77 (−1.12, −0.41)	−4.61	0.0003
PC aa C40:6	−0.74 (−1.05, −0.44)	−5.05	0.00005	PC ae C34:2	−0.84 (−1.14, −0.54)	−6.04	0.00002
PC ae C30:2	−0.85 (−1.09, −0.61)	−7.42	<0.00001	PC ae C36:2	−0.85 (−1.14, −0.56)	−6.24	0.00002
PC ae C32:1	−0.80 (−1.07, −0.53)	−6.09	<0.00001	PC ae C36:5	−0.75 (−1.11, −0.38)	−4.33	0.0006
PC ae C34:0	−0.81 (−1.08, −0.55)	−6.35	<0.00001	PC ae C38:2	−0.80 (−1.13, −0.48)	−5.25	0.0001
PC ae C34:2	−0.74 (−1.04, −0.43)	−5.03	0.00006	PC ae C38:6	−0.85 (−1.14, −0.56)	−6.31	0.00001
PC ae C34:3	−0.88 (−1.10, −0.66)	−8.50	<0.00001	PC ae C40:6	−0.82 (−1.13, −0.50)	−5.50	0.00006
PC ae C36:1	−0.80 (−1.07, −0.53)	−6.10	<0.00001	SM (OH) C14:1	−0.87 (−1.14, −0.59)	−6.76	<0.00001
PC ae C36:2	−0.92 (−1.10, −0.74)	−10.61	<0.00001	SM (OH) C16:1	−0.85 (−1.14, −0.57)	−6.32	0.00001
PC ae C38:2	−0.92 (−1.10, −0.75)	−11.09	<0.00001	SM (OH) C22:1	−0.81 (−1.13, −0.49)	−5.37	0.00008
PC ae C38:3	−0.83 (−1.08, −0.58)	−6.85	<0.00001	SM (OH) C22:2	−0.85 (−1.14, −0.55)	−6.12	0.00002
PC ae C38:4	−0.87 (−1.09, −0.65)	−8.07	<0.00001	SM C16:0	−0.74 (−1.11, −0.38)	−4.31	0.0006
PC ae C38:6	−0.87 (−1.09, −0.65)	−8.14	<0.00001	SM C16:1	−0.83 (−1.14, −0.52)	−5.72	0.00004
PC ae C40:2	−0.84 (−1.09, −0.60)	−7.20	<0.00001	SM C20:2	0.81 (0.49, 1.13)	5.42	0.00007
PC ae C40:4	−0.78 (−1.07, −0.50)	−5.77	0.00001				
PC ae C40:5	−0.82 (−1.08, −0.56)	−6.63	<0.00001				
PC ae C40:6	−0.87 (−1.09, −0.65)	−8.16	<0.00001				
PC ae C42:5	−0.82 (−1.08, −0.56)	−6.56	<0.00001				
SM (OH) C14:1	−0.90 (−1.10, −0.70)	−9.26	<0.00001				
SM (OH) C16:1	−0.85 (−1.09, −0.60)	−7.25	<0.00001				
SM (OH) C22:1	−0.92 (−1.10, −0.74)	−10.55	<0.00001				
SM (OH) C22:2	−0.92 (−1.10, −0.75)	−11.04	<0.00001				
SM C16:0	−0.86 (−1.09, −0.63)	−7.80	<0.00001				
SM C16:1	−0.90 (−1.10, −0.69)	−9.23	<0.00001				
SM C24:0	−0.81 (−1.07, −0.55)	−6.40	<0.00001				
SM C24:1	−0.85 (−1.09, −0.62)	−7.52	<0.00001				

Statistically significant regression coefficients (ß), confidence intervals (CI), and *t*- and *p*-values (derived from GLM analysis) of log2-transformed variables in standard diet and HFD groups.

**Table 2 ijms-23-11682-t002:** Effect of HFD on metabolite levels among Bl6 and 129Sv mice.

Bl6				129Sv			
	ß (ß 95% CI)	*t*-Value	*p*-Value		ß (ß 95% CI)	*t*-Value	*p*-Value
C0	0.91 (0.72, 1.10)	9.90	<0.00001	9-week weight gain (%)	−0.60 (−1.02, −0.18)	−3.01	0.008
C2	0.89 (0.69, 1.10)	8.94	<0.00001	C0	0.94 (0.76, 1.12)	11.14	<0.00001
C4-	0.70 (0.36, 1.03)	4.36	0.0003	C2	0.85 (0.57, 1.13)	6.45	<0.00001
C18	−0.77 (−1.07, −0.47)	−5.34	0.00003	C3	0.68 (0.30, 1.07)	3.74	0.002
C18:2	0.75 (0.43, 1.06)	5.00	0.00007	C4-	0.89 (0.65, 1.13)	7.90	<0.00001
Kynurenine	0.73 (0.41, 1.05)	4.73	0.0001	C14:2	0.77 (0.43, 1.11)	4.77	0.0002
lysoPC a C18:0	−0.78 (−1.07, −0.48)	−5.52	0.00002	C18:2	0.78 (0.45, 1.11)	5.04	0.0001
lysoPC a C18:1	−0.88 (−1.10, −0.66)	−8.40	<0.00001	Gly	0.83 (0.53, 1.12)	5.85	0.00002
lysoPC a C20:3	−0.89 (−1.10, −0.67)	−8.58	<0.00001	Kynurenine	0.64 (0.23, 1.05)	3.34	0.004
lysoPC a C26:0	−0.70 (−1.03, −0.36)	−4.34	0.0003	t4-OH-Pro	0.76 (0.41, 1.10)	4.63	0.0003
PC aa C28:1	−0.83 (−1.09, −0.57)	−6.71	<0.00001	lysoPC a C18:0	−0.75 (−1.10, −0.40)	−4.58	0.0003
PC aa C30:2	−0.88 (−1.10, −0.66)	−8.43	<0.00001	lysoPC a C18:1	−0.88 (−1.13, −0.63)	−7.49	<0.00001
PC aa C32:1	−0.81 (−1.08, −0.53)	−6.14	<0.00001	lysoPC a C20:3	−0.94 (−1.12, −0.75)	−10.60	<0.00001
PC aa C34:1	−0.96 (−1.09, −0.82)	−14.97	<0.00001	lysoPC a C26:0	−0.58 (−1.01, −0.15)	−2.88	0.01
PC aa C34:2	−0.76 (−1.07, −0.46)	−5.30	0.00003	lysoPC a C28:1	−0.82 (−1.12, −0.53)	−5.84	0.00003
PC aa C36:1	−0.97 (−1.08, −0.86)	−17.93	<0.00001	PC aa C28:1	−0.89 (−1.13, −0.65)	−7.87	<0.00001
PC aa C36:2	−0.95 (−1.09, −0.82)	−14.31	<0.00001	PC aa C30:2	−0.93 (−1.13, −0.73)	−9.97	<0.00001
PC aa C36:3	−0.94 (−1.10, −0.78)	−12.20	<0.00001	PC aa C32:3	−0.80 (−1.12, −0.48)	−5.33	0.00007
PC aa C38:0	−0.71 (−1.04, −0.38)	−4.49	0.0002	PC aa C34:1	−0.86 (−1.13, −0.60)	−6.85	<0.00001
PC aa C38:3	−0.92 (−1.10, −0.75)	−10.87	<0.00001	PC aa C36:1	−0.96 (−1.11, −0.80)	−13.06	<0.00001
PC aa C38:4	−0.68 (−1.02, −0.34)	−4.19	0.0005	PC aa C36:2	−0.94 (−1.12, −0.77)	−11.50	<0.00001
PC aa C38:5	−0.67 (−1.01, −0.32)	−4.00	0.0007	PC aa C36:3	−0.92 (−1.13, −0.71)	−9.43	<0.00001
PC aa C40:3	−0.69 (−1.03, −0.36)	−4.30	0.0004	PC aa C38:1	−0.77 (−1.11, −0.44)	−4.87	0.0002
PC ae C30:2	−0.79 (−1.08, −0.51)	−5.81	0.00001	PC aa C38:3	−0.92 (−1.13, −0.72)	−9.73	<0.00001
PC ae C32:1	−0.76 (−1.06, −0.45)	−5.17	0.00005	PC aa C38:5	−0.77 (−1.11, −0.44)	−4.87	0.0002
PC ae C32:2	−0.91 (−1.10, −0.72)	−10.08	<0.00001	PC aa C40:3	−0.62 (−1.03, −0.20)	−3.12	0.007
PC ae C34:1	−0.95 (−1.10, −0.80)	−13.15	<0.00001	PC aa C40:5	−0.81 (−1.12, −0.50)	−5.55	0.00004
PC ae C34:3	−0.80 (−1.08, −0.52)	−5.98	<0.00001	PC ae C30:2	0.84 (0.56, 1.13)	6.24	0.00001
PC ae C36:0	−0.84 (−1.09, −0.59)	−6.97	<0.00001	PC ae C32:1	−0.72 (−1.09, −0.35)	−4.10	0.0008
PC ae C36:1	−0.96 (−1.09, −0.84)	−16.05	<0.00001	PC ae C32:2	−0.87 (−1.13, −0.62)	−7.18	<0.00001
PC ae C36:2	−0.73 (−1.05, −0.41)	−4.79	0.0001	PC ae C34:1	−0.87 (−1.13, −0.60)	−6.92	<0.00001
PC ae C36:3	−0.89 (−1.10, −0.68)	−8.85	<0.00001	PC ae C34:3	−0.73 (−1.09, −0.37)	−4.30	0.0006
PC ae C36:4	−0.71 (−1.04, −0.38)	−4.48	0.0002	PC ae C36:0	−0.74 (−1.10, −0.39)	−4.42	0.0004
PC ae C38:1	−0.83 (−1.09, −0.57)	−6.58	<0.00001	PC ae C36:1	−0.86 (−1.13, −0.58)	−6.65	<0.00001
PC ae C38:3	−0.85 (−1.10, −0.61)	−7.31	<0.00001	PC ae C36:3	−0.75 (−1.10, −0.40)	−4.58	0.0003
PC ae C38:4	−0.67 (−1.01, −0.32)	−4.00	0.0007	PC ae C36:5	−0.74 (−1.10, −0.38)	−4.39	0.0005
PC ae C38:5	−0.91 (−1.10, −0.71)	−9.74	<0.00001	PC ae C38:1	−0.79 (−1.11, −0.46)	−5.15	0.0001
PC ae C40:5	−0.71 (−1.04, −0.38)	−4.50	0.0002	PC ae C38:2	0.91 (0.69, 1.13)	8.81	<0.00001
PC ae C42:2	−0.81 (−1.08, −0.54)	−6.16	<0.00001	PC ae C38:3	−0.83 (−1.13, −0.53)	−5.95	0.00002
SM (OH) C14:1	−0.83 (−1.09, −0.57)	−6.72	<0.00001	PC ae C38:5	−0.88 (−1.13, −0.63)	−7.40	<0.00001
SM (OH) C16:1	−0.71 (−1.04, −0.39)	−4.54	0.0002	PC ae C40:4	0.88 (0.62, 1.13)	7.26	<0.00001
SM C16:0	−0,92 (−1.10, −0.74)	−10.49	<0.00001	PC ae C42:0	0.77 (0.43, 1.11)	4.84	0.0002
SM C16:1	−0.80 (−1.08, −0.52)	−6.00	<0.00001	PC ae C42:2	−0.76 (−1.10, −0.42)	−4.68	0.0003
SM C18:0	−0,87 (−1.10, −0.64)	−7.81	<0.00001	SM (OH) C14:1	−0.90 (−1.13, −0.67)	−8.24	<0.00001
SM C18:1	−0.70 (−1.04, −0.37)	−4.44	0.0002	SM (OH) C16:1	−0.88 (−1.13, −0.63)	−7.52	<0.00001
SM C22:3	−0.66 (−1.01, −0.31)	−3.95	0.0008	SM (OH) C22:2	0.89 (0.65, 1.13)	7.89	<0.00001
SM C24:0	−0.75 (−1.06, −0.44)	−5.06	0.00006	SM C16:0	−0.94 (−1.12, −0.76)	−11.14	<0.00001
				SM C16:1	−0.76 (−1.10, −0.41)	−4.65	0.0003
				SM C18:1	−0.78 (−1.11, −0.44)	−4.94	0.0001

Statistically significant regression coefficients (ß), confidence intervals (CI), and t- and *p*-values (derived from GLM analysis) of log2-transformed metabolite levels.

**Table 3 ijms-23-11682-t003:** Diet composition.

	Control Diet (3.230 kcal/kg)	High-Fat Diet (4.615 kcal/kg)
Fat (%)	9 kJ	45 kJ
Protein (%)	24 kJ	20 kJ
Carbohydrate (%)	67 kJ	35 kJ
	**Crude Nutrients [%]**	
Crude protein (N × 6.25)	19.0	22.0
Crude fat	3.3	23.6
Crude fiber	4.9	5.7
Crude ash	6.4	5.3
Starch	35.2	6.8
Sugar	5.3	21.1
N free extracts	54.2	40.0

## Data Availability

The data presented in this study are available on request from the corresponding author.

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
