# Peer review of "Impact of a High-Fat Diet on the Metabolomics Profile of 129S6 and C57BL6 Mouse Strains"

_ijms, 2022, doi:10.3390/ijms231911682_

Round 1

Reviewer 1 Report

In the reviewed manuscript, written by Maria Piirsalu and colleagues, the authors studied the impact of a 9- weeks high-fat diet consumption on the metabolomics profile of 129Sv 2 and Bl6 mouse strains. The manuscript is interesting and provides many results that allow to compare 2 strains of mice, both in the context of research focused on nutrition, obesity, and pharmacology/pharmacokinetics. Below are a few inquiries:

  • Why did the authors only study males? More and more studies are also involving female rodents. It would be interesting to study the sexual dimorphism response to exposure to HFD.

  • Did the authors notice any changes in organ size /weight in animals consuming HFD?

  • A diet with a fat content of 45%, and a period of consumption of this diet for 9 weeks was selected for the research. Other studies using a 60% fat diet can also be found in the literature. Could different dietary fat content (higher or lower) lead to different results? How does this translate into the translational nature of research? 

  • The open field test is a generally accepted paradigm for analyzing exploratory and motor behavior in rodents. It enables the assessment of reactions to new and unknown environments and environmental habituation. To test the anxiety-like behavior, additional verification tests, e.g. elevated zero/plus maze, could be performed. Also in the context of characterizing mouse strains, it would be interesting to compare the social interactions.

  • Editorial comments:
  • It would be helpful for the reader to add a percentage to the pie chart in Figure 1K.
  • Figure 2 - legend indicating HFD or control group is missing (part A-K).

Author Response

We appreciate the time and effort dedicated to providing feedback and thank you for your thoughtful comments and efforts towards improving our manuscript. Please see below, in bold, for a point-by-point response to the comments.

  • Why did the authors only study males? More and more studies are also involving female rodents. It would be interesting to study the sexual dimorphism response to exposure to HFD.
I completely agree with the reviewer that it would be interesting study the sexual dimorphism response to HFD.  Male mice are typically chosen for animal studies in the field of metabolic disorders and obesity because they exhibit more pronounced obesity-related phenotypes than female mice. Male mice are usually much bigger and they gain significantly more in body weight. This feature enables better distinction of diet induced changes. In addition, the metabolic variability caused by the estrous cycle in female mice is not present in male mice. Most importantly, in our previous study (where we utilized both sexes), we saw a more robust HFD induced phenotype in male mice. These animals were in a mixed Bl6/129Sv background.    Kaare, M.; Mikheim, K.; Lilleväli, K.; Kilk, K.; Jagomäe, T.; Leidmaa, E.; Piirsalu, M.; Porosk, R.; Singh, K.; Reimets, R.; Taalberg, E.; Schäfer, M.K.E.; Plaas, M.; Vasar, E.; Philips, M.-A. High-Fat Diet Induces Pre-Diabetes and Distinct Sex-Specific Metabolic Alterations in Negr1-Deficient Mice. Biomedicines 2021, 9, 1148. https://doi.org/10.3390/biomedicines9091148  
  • Did the authors notice any changes in organ size /weight in animals consuming HFD?
Indeed, it would have been an interesting addition to the paper! Unfortunately we did not examine the size and weight of organs. 
  • A diet with a fat content of 45%, and a period of consumption of this diet for 9 weeks was selected for the research. Other studies using a 60% fat diet can also be found in the literature. Could different dietary fat content (higher or lower) lead to different results? How does this translate into the translational nature of research? 
Very interesting question. Surely different dietary fat content may have a different outcome. However, this was not the purpose of this study. The diet was chosen based on our previous research, where we have shown that the HFD with 45% fat content induced distinct sex-specific metabolic alterations in Negr1-deficient mice (Kaare et al., 2021).  The manufacturers of this diet promote it as a "western" diet, which has a catastrophic effect on human metabolism and health, which we can see also in this study (Wilson et. al., 2007). In addition, the use of different diets would have made the number of experimental animals in groups too high, which would have not been in line with the 3R principle.   Kaare, M.; Mikheim, K.; Lilleväli, K.; Kilk, K.; Jagomäe, T.; Leidmaa, E.; Piirsalu, M.; Porosk, R.; Singh, K.; Reimets, R.; Taalberg, E.; Schäfer, M.K.E.; Plaas, M.; Vasar, E.; Philips, M.-A. High-Fat Diet Induces Pre-Diabetes and Distinct Sex-Specific Metabolic Alterations in Negr1-Deficient Mice. Biomedicines 2021, 9, 1148. https://doi.org/10.3390/biomedicines9091148   Wilson CR, Tran MK, Salazar KL, Young ME, Taegtmeyer H. Western diet, but not high fat diet, causes derangements of fatty acid metabolism and contractile dysfunction in the heart of Wistar rats. Biochem J. 2007 Sep 15;406(3):457-67. doi: 10.1042/BJ20070392. PMID: 17550347; PMCID: PMC2049036.
  • The open field test is a generally accepted paradigm for analyzing exploratory and motor behavior in rodents. It enables the assessment of reactions to new and unknown environments and environmental habituation. To test the anxiety-like behavior, additional verification tests, e.g. elevated zero/plus maze, could be performed. Also in the context of characterizing mouse strains, it would be interesting to compare the social interactions. 
Thank you for the suggestion! Although we agree that additional tests would have been interesting, we believe that excess handling of mice would have affected metabolic outcome. We chose the Phenotyper open field test because it allows us to examine overall activity, anxiety-related, and exploratory behaviors in a single test. Since we studied the metabolic profile, it is important to avoid excess handling of the animals because behavioral experiments can have an impact on the metabolic profile. Considering that these two mouse lines differ in their behavioral repertoire and that behavioral experiments may have an effect on the metabolic profile, we decided to use only one methodology to characterize behavior. In addition, in our previous experiments, we saw a strong dissonance during the first two hours of Phenotyper testing. The locomotor activity of 129Sv was suppressed during the first two hours, most likely reflecting a higher anxiety-like behavior of 129Sv at the beginning of the experiment and a passive adaptation to a stressful environment (Piirsalu et al., 2020). Therefore, we decided to use the same method this time.  With Phenotyper, we can observe animal behavior long enough to see their activity in the dark and light phases. In addition, we can divide the observation time into hourly values, which allows us to see the behavioral dynamics over the established time. Phenotyper also allows us to virtually divide the open field arena into central and peripheral zones, as well as water concumption and feeding zones. We have added the first 3 h locomotor data into the supplementary material and to the results (Line 154-186) and discussion (Line 400-416) of the manuscript. We would also like to emphasize that this work is part of a larger study evaluating validity of 129Sv for modelling of neuropsychiatric conditions.    Piirsalu, M.; Taalberg, E.; Lilleväli, K.; Tian, L.; Zilmer, M.; Vasar, E. Treatment With Lipopolysaccharide Induces Distinct Changes in Metabolite Profile and Body Weight in 129Sv and Bl6 Mouse Strains. Front Pharmacol 2020, 11, 371, doi:10.3389/fphar.2020.00371.  
  • Editorial comments:
  • It would be helpful for the reader to add a percentage to the pie chart in Figure 1K.
  • Figure 2 - legend indicating HFD or control group is missing (part A-K).
Thank you for these suggestions. The percentages were added to the pie chart in Figure 1K and legend added to Figure 2.

Reviewer 2 Report

I fail to be able to understand a significant and novel aspect as well as the the rationale for the large body of work.  The simple comparisons of two mouse strains has value in the data provided but  fails to present a novel and compelling aspect or publication

Author Response

Thank you for your time and work in revising our paper. Please see below, in bold, our response to the concerns. 

I fail to be able to understand a significant and novel aspect as well as the the rationale for the large body of work. The simple comparisons of two mouse strains has value in the data provided but fails to present a novel and compelling aspect or publication   We very much appreciate the reviewer's feedback, but would like to disagree. We would like to believe that this study makes a necessary contribution to the field of neuroscience. Indeed, the Bl6 line is widely used in biomedical research. It is the predominant inbred mouse strain in animal studies. However, it cannot be the best choice for all kinds of studies. The aim of our research is to show that the simultaneous use of 129Sv, neglected in many cases, helps to overcome the obstacles inherent in the use of only the Bl6 line. Mice are used as experimental animals for modeling a variety of human diseases. In the application of transgenic technology, the Bl6 line is clearly dominating. However, it is worth noting that the 129Sv line has played an equal role in the initial application of transgenic technology. For translational research, it is critical that animal models are accurately characterized and validated as models of human disease. In mice, and especially in genetically modified models, special considerations must be made because these modifications are influenced by the genetics of the background strain, husbandry, and experimental conditions. Thus, it is very important to understand the translational value of these different mouse lines and to find out for which studies of human pathology one or the other line may be appropriate for. Our studies show that active adaptation responses predominate in Bl6 mice and passive adaptation predominates in 129Sv mice. Our previous studies show that 129Sv mice are more suitable for studying anxiety-, depression-, and psychosis-like states. At the same time, the Bl6 line is preferred when studying social dominance, aggressiveness, addictive behaviors, and conditions that require rapid adaptation. In the case of psychiatric diseases, such as schizophrenia spectrum disorders, major depressive disorder and bipolar disorder, the development of obesity and metabolic syndrome occurs in patients parallel to the development of the disease, which reduces the effectiveness of treatment and complicates the further course of the disease. In the present work, we found that under established conditions, the 129Sv mouse line exhibits significant weight gain on HFD with the similar metabolite shifts seen in humans with metabolic syndrome. In addition, we have found that male 129Sv mice tend to gain significantly more weight on a normal diet. Therefore, 129Sv mice can be considered a good model for studying the metabolic syndrome associated with psychiatric disorders. By using Bl6 mice in parallel, it is possible to elucidate the mechanisms that provide protection against these alterations. Thus, the novelty of this research is in establishing the necessary connection between metabolic syndrome and psychiatric disorder-like conditions using 129Sv mice. We have rewritten the conclusions section to underline the translational value of our research.

Reviewer 3 Report

The manuscript by Piirsalu et al. is very interesting and describes similarities and differences in the body weight change, behavior, and several metabolic variables in two mouse strains -  Bl6 and 129Sv - in response to high-fat diet (HFD). The manuscript is well prepared but I suggest some corrections and improvements. 

1. The introduction is well written, but add a reference in the first paragraph.

2. Line 46 - There is dot which is redundant. Correct it.

3. Line 74 – HFD - write the full name when the abbreviation first appears.

4. Line 109 - write the p value in italics

5. Figure 1 - write a better name for this picture. Također, add the scattergram of correlative relationship between average food intake and mass change of animals per day

6. Line 117 - **p ≤ 0.01 - write in comparation to what is statistically significant?!

7. Enlarge the Figure 2 Z because it is very difficult to see. Also, in the description of Figure 2 vs write in italics. Please correct it.

8. Line 171 - *p ≤ 0.05 - write in comparation to what is statistically significant.

9. Line 187 – 24h - here you have written the hours with the number, and above in the text it is written separately. Please make it uniform throughout the work.

10. 2.4. Metabolic changes induced by high fat diet (HFD) - the full HFD name is unnecessary here.

11. Figure 4 - Use different symbols to indicate statistical significance in different groups. Revise it throughout the paper when displaying the results

12. Lines 349 – 350 - We have repeatedly shown that metabolites C4-, C5- and SM(OH) 349 C22:2 are significantly higher in 129Sv mice and carnosine, alpha-AAA, Ac-Orn and 350 lysoPC a C16:1 belong to the metabolic signature of Bl6 mice - excess space in the text. Pass spaces through the entire text.

13. Line 390 – vice versa – write it in italics because it is Latin name.

14. Paragraph 3.5. there is no explanation for the obtained results or confirmation. Deepen this paragraph.

15. Why were 28 B16 mice and 30 129Sv mice used in the experiment? Why is that number not the same in both strains?

16. Table 3 – diet composition – correct it

17. There is no reference for HPLC when measuring metabolites. The method is not sufficiently described. Describe the column, the gradient program, and indicate the amount of injected sample.

18. The conclusion is too long. It describes the results instead of briefly presenting the conclusions from the obtained results. Revise the conclusion and rewrite it.

19. Lines 634-335 - Delete this sentence. This is for discussion.

20. References are not written according to the instructions of the Journal. Rewrite it.

Author Response

We appreciate the time and effort dedicated to providing feedback and thank you for your thorough comments and efforts towards improving our manuscript. We have incorporated most of the suggestions. Please see below, in bold, for a point-by-point response to the comments and concerns. 

  •  The introduction is well written, but add a reference in the first paragraph.
Thank you for pointing this out! Reference has been added to the first paragraph of introduction.  
  • Line 46 - There is dot which is redundant. Correct it.
Thank you! Correction has been made.  
  •  Line 74 – HFD - write the full name when the abbreviation first appears.
The abbreviation has been corrected.  
  • Line 109 - write the p value in italics
Correction has been made.  
  • Figure 1 - write a better name for this picture. TakoÄ‘er, add the scattergram of correlative relationship between average food intake and mass change of animals per day
Thank you for this suggestion! Unfortunately, we cannot add the scatter plot of the correlative relationship between average food intake and animal weight change per day because to do so, we would have had to isolate mice to single cages. However, social isolation would have had an unfavorable effect on metabolic profiling. Food intake was recorded weekly by subtracting the mass (g) of remaining food from the initial amount of food provided to each cage of 4-5 mice.  As suggested, we have changed the name of Figure 1.  
  • Line 117 - **p ≤ 0.01 - write in comparison to what is statistically significant?!
Thank you! Correction has been made.  
  • Enlarge the Figure 2 Z because it is very difficult to see. Also, in the description of Figure 2 vs write in italics. Please correct it.
The figure 2Z has been corrected.  
  • Line 171 - *p ≤ 0.05 - write in comparison to what is statistically significant.
Thank you for this comment! We have made appropriate corrections.  
  • Line 187 – 24h - here you have written the hours with the number, and above in the text it is written separately. Please make it uniform throughout the work.
Thank you very much for this comment! The corrections were made to improve the consistency of the work.  
  • 2.4. Metabolic changes induced by high fat diet (HFD) - the full HFD name is unnecessary here.
Thank you for this comment! We have made appropriate corrections.  
  • Figure 4 - Use different symbols to indicate statistical significance in different groups. Revise it throughout the paper when displaying the results.
Thank you very much for this suggestion! The symbols indicating statistical significance in different groups have been adjusted throughout the paper.  
  • Lines 349 – 350 - We have repeatedly shown that metabolites C4-, C5- and SM(OH) 349 C22:2 are significantly higher in 129Sv mice and carnosine, alpha-AAA, Ac-Orn and 350 lysoPC a C16:1 belong to the metabolic signature of Bl6 mice - excess space in the text. Pass spaces through the entire text.
Thank you for noticing this! We have removed the excess space.   
  • Line 390 – vice versa – write it in italics because it is Latin name.
Thank you for pointing this out. Appropriate corrections have been made.   
  • Paragraph 3.5. there is no explanation for the obtained results or confirmation. Deepen this paragraph.
Thank you for pointing this out. We have expanded paragraph 3.5.  
  • Why were 28 B16 mice and 30 129Sv mice used in the experiment? Why is that number not the same in both strains?
We took the maximum number of available experimental animals provided by the animal facility to increase statistical power without conflicting with the 3Rs principle.  
  • Table 3 – diet composition – correct it
As suggested by the reviewer, we have corrected the table title.  
  • There is no reference for HPLC when measuring metabolites. The method is not sufficiently described. Describe the column, the gradient program, and indicate the amount of injected sample.
Thank you very much for pointing this out. We have provided a more detailed description for metabolite measurement.  
  • The conclusion is too long. It describes the results instead of briefly presenting the conclusions from the obtained results. Revise the conclusion and rewrite it.
Thank you for this comment! We have rewritten the conclusions section.  
  • Lines 634-335 - Delete this sentence. This is for discussion.
This sentence has been removed from the conclusions section.  
  • References are not written according to the instructions of the Journal. Rewrite it.
Thank you for pointing this out. The reviewer is correct, and we have rewritten the references according to the IJMS instructions.